# The Significance of NLRP Inflammasome in Neuropsychiatric Disorders

**DOI:** 10.3390/brainsci12081057

**Published:** 2022-08-10

**Authors:** Yao Shen, Liyin Qian, Hu Luo, Xiaofang Li, Yuer Ruan, Runyue Fan, Zizhen Si, Yunpeng Chen, Longhui Li, Yu Liu

**Affiliations:** 1Department of Public Health, School of Medicine, Ningbo University, Ningbo 315021, China; 2Department of Psychology, Faculty of Teacher Education, Ningbo University, Ningbo 315021, China; 3Ningbo Yinzhou District Center for Disease Control and Prevention, Ningbo 315199, China; 4Department of Physiological Pharmacology, School of Medicine, Ningbo University, Ningbo 315021, China; 5Department of Pharmacology, Affiliated Hospital of Medical School, Ningbo University, Ningbo 315020, China; 6Ningbo Kangning Hospital, Ningbo 315201, China

**Keywords:** NLRP inflammasome, neuropsychiatric disorders, neuroinflammation, pharmacological treatment

## Abstract

The NLRP inflammasome is a multi-protein complex which mainly consists of the nucleotide-binding oligomerization domain, leucine-rich repeat, and pyrin domain. Its activation is linked to microglial-mediated neuroinflammation and partial neuronal degeneration. Many neuropsychiatric illnesses have increased inflammatory responses as both a primary cause and a defining feature. The NLRP inflammasome inhibition delays the progression and alleviates the deteriorating effects of neuroinflammation on several neuropsychiatric disorders. Evidence on the central effects of the NLRP inflammasome potentially provides the scientific base of a promising drug target for the treatment of neuropsychiatric disorders. This review elucidates the classification, composition, and functions of the NLRP inflammasomes. It also explores the underlying mechanisms of NLRP inflammasome activation and its divergent role in neuropsychiatric disorders, including Alzheimer’s disease, Huntington’s disease, Parkinson’s disease, depression, drug use disorders, and anxiety. Furthermore, we explore the treatment potential of the NLRP inflammasome inhibitors against these disorders.

## 1. Introduction

Neuropsychiatric disorders have a significant impact on human health and quality of life, causing a huge socio-economic burden on society and overstretching the healthcare system. Therefore, studies investigating the mechanisms of neuropsychiatric disorders for effective therapies have increased. Activated inflammatory responses are a major cause and common feature of numerous neuropsychiatric disorders [1,2]. Nucleotide-binding oligomerization domain (NOD) and leucine-rich repeat (LRR)-containing receptors or NOD-like receptors (NLRs) are inflammasomes that are critical to initiate innate immune responses to host-derived danger signals [3]. Activations of many prototypic NLRs, including a NLR with a pyrin domain (NLRP) containing NLRP1, NLRP3, and NLRP4, result in the maturation and release of different pro-inflammatory cytokines (IL-1β and IL-18) [4]. The process has been suggested to be of great importance to the occurrence of programmed cell death, which is called pyroptosis [5]. As a part of the innate immune system, the NLRP inflammasome regulates the host’s defense against harmful threats. Its activation is implicated in microglial-mediated neuroinflammation and partial neuronal degeneration [6]. In addition, it modulates the pathogenesis of various neuropsychiatric disorders, including Alzheimer’s disease, Huntington’s disease, Parkinson’s disease, Amyotrophic lateral sclerosis (ALS), traumatic brain injury (TBI), drug use disorder, depression, and anxiety [7]. This review explores the significance and therapeutical implications of NLRP inflammasomes in neuropsychiatric disorders, geared towards providing a basis for exploring common pathway-based treatments for different neuropsychiatric diseases.

## 2. NLRP Inflammasome

NLRP inflammasome is a three-part multi-protein complex which senses danger signals via the nucleotide-binding oligomerization domain (NOD) such as receptors (NLR) containing a CARD (C-terminal caspase-recruitment domain), and controls the activation of Caspase (Casp-1) [8]. The NLRP inflammasome family consists of more than 20 species, including NLRP1, NLRP3, and NLRP4. Studies on neuropsychiatric disorders primarily focus on NLRP1 and NLRP3 inflammasome [9]. NLRP1 inflammasome, majorly expressed in the microglia and neurons of the brain, was the first member of the NLRP family to be identified [10]. This inflammasome comprises a receptor protein (NLRP1), an adaptor protein (ASC), and an effector protein (pro-caspase-1) [11]. NLRP3 inflammasome, primarily localized in the microglia, was the first inflammasome to be investigated in the brain. NLRP3 inflammasome is composed of NLRP3, ASC, and pro-caspase-1. As depicted in Figure 1, the structure of the NLRP inflammasome includes ASC, NLRPs, pro-caspase-1. The adaptor ASC has two protein interaction domains, an N-terminal PYD and a CARD [12]. Most inflammatory vesicles are activated by only one or a few highly specific agonists, but NLRP1 and NLRP3 can be activated by many agonists with a Toll-like receptor agonist (lipopolysaccharide (LPS), nigericin, monosodium urate crystals, and adenosine-triphosphate), pathogens (bacteria, fungi and viruses), or a proinflammatory cytokine (tumor necrosis factor, TNF) [13]. It can also be activated by various signalings specifically associated with neuropsychiatric disorders, including K^+^ or Cl^−^, Ca^2+^, lysosomal disruption, mitochondrial dysfunction, metabolic changes and trans-Golgi catabolism [12,14]. In microglia, NLRP3 inflammasome becomes activated when these cells sense proteins such as misfolded or aggregated amyloid-β, α-synuclein and prion protein or superoxide dismutase, ATP and members of the complement pathway, and results in the maturation and release of various pro-inflammatory cytokines (e.g., IL-1β, and IL-18) [7]. IL-1β promotes inflammatory responses including leukocyte infiltrations, lymphocyte activation and acute phase protein induction. Moreover, after binding to IL-1β receptors, it induces the secretion of large amounts of inflammatory factors and chemokines [15,16]. IL-18 exerts its pro-inflammatory effects by stimulating the production of nitric oxide and reactive oxygen species. NLRP3 inflammasome activation results in caspase-1 activation, in turn causing cleavage of pro-IL-1β and gasdermin D (GSDMD). Active GSDMD aggregates and forms pores in the cell membrane, resulting in cell swelling and a programmed cell death which is called pyroptosis [17]. Pyroptosis is implicated in the pathogenesis of several neuropsychiatric disorders, such as multiple sclerosis (MS), Alzheimer’s disease, TBI, drug use disorder, and depression [18,19]. In addition to microglia, NLRP1 and NLRP3 can also be found in myeloid cells in the central nervous system and may also contribute to the modulation of central innate immunity [20,21].

The mechanisms leading to NLRP inflammasome activation are intensely debated. Some of the detailed signaling pathways involved in NLRP1/NLRP3 inflammasome activation are described in this article, such as elevated ROS, K^+^/Ca^2+^ imbalance and autophagic inhibition in activating NLRP inflammasome in neuropsychiatric disorders. The mechanisms of action in NLRP inflammasomes in neuropsychiatric disorders are mainly depicted in Figure 2. Recent reports have revealed complex interactions between the inflammasome and ROS pathways. Chronic cerebral hypoperfusion (CCH) induces ROS accumulation and promotes the activation of NLRP3 inflammasomes and the release of IL-1β. However, URB597 (URB) alleviated autophagy and mitochondrial impairment by reducing the mitochondrial ROS as well as restoring the lysosomal function, which further inhibited the NLRP3-CASP1 pathway activation in the rat hippocampus [22]. Acetoxychavicol acetate (ACA) inhibited NLRP3 agonists (e.g., nigericin, MSU crystals, and ATP) in mouse bone marrow-derived macrophages and NLRP3 inflammasome activation in human THP-1 monocytes by suppressing the production of mitochondrial reactive oxygen species (ROS) [5]. In addition, it inhibited the oligomerization of adapter molecules, ASC and the cleavage of the cystein-1 mediated pyroptosis actuator Gasdermin D [5]. K^+^/Ca^2+^ imbalance plays an important role in the activation of NLRP3 inflammasomes. Ca^2+^ influx and K^+^ efflux promote NLRP3 inflammasome activation in mice. In addition, aldose reductase (AR) regulates NLRP3 inflammation-mediated innate immune responses by altering the ROS/LYN/SYK/PI3K/Ca^2+^/K^+^ signaling pathway [23]. Contact of the P2X7 purinoceptor with extracellular ATP induces transmembrane K^+^/Ca^2+^ imbalance, leading to activation of NLRP1 and NLRP3 inflammasomes in LPS-stimulated macrophages. This evidence suggests that both potassium efflux and calcium influx are necessary for the generation of mitochondrial ROS and to trigger NLRP inflammation [24]. Autophagic activity is maintained at relatively low levels under steady-state conditions, but is effectively induced by various cellular stresses, such as organelle damage and pathogen infection [25]. Recent studies have shown that autophagy, an intracellular degradation system associated with the maintenance of cellular homeostasis, plays a key role in the inactivation of the inflammasome. Notably, autophagy deficiency caused by genetic mutations can disrupt organelle elimination, thereby inducing aberrant activation of the inflammasome and leading to severe tissue damage [26]. Blocked autophagy and mitochondrial flux also enhanced the activation of NLRP3-CASP1 pathways. Autophagy inhibition can lead to lysosomal damage, resulting in the cytoplasmic release of lysosomal contents which (e.g.,) activate NLRP inflammasome [27]. Mechanistically, Kaempferol (KA) promotes macrophage/autophagy in microglia and promotes neuroinflammatory suppressive effects through the cooperation of ubiquitination and autophagy, leading to the reduced expression of NLRP3 proteins and consequently to the deactivation of NLRP3 inflammasomes in mice [28]. The blockade of autophagy by genetic ablation of the autophagy regulators Atg16L1 or Atg7 makes LPS-dependent inflammasome activation possible in the central nervous system (CNS) in mice [29]. In conclusion, NLRP inflammasome activators play an important role in the activation of inflammatory vesicles by triggering multiple cellular and molecular events including potassium-calcium ion imbalance, mitochondrial dysfunction and lysosomal rupture, especially alterations in intracellular ion levels that link different signal transductions; however, the detailed mechanisms of inflammatory vesicle activation by each activation signal still deserve more in-depth investigation. Information about the chemical structure of NLRP inflammatory vesicle inhibitors, models of drug treatment of disease, and clinical advances in drug treatment of neuropsychiatric disorders are described mainly in Table 1.

## 3. Roles of NLRP Inflammasome in Neuropsychiatric Disorders

Neuroinflammation is a vital factor in the pathogenesis of psychiatric illnesses, including Alzheimer’s disease, Huntington’s disease, Parkinson’s disease, drug use disorders, depression, and anxiety. In addition, it is involved in sickness behaviors, diminished cognition, and memory [38].

### 3.1. Alzheimer’s Disease 

Alzheimer’s disease is a neurodegenerative disease with the characteristics of memory loss and cognitive decreases. It is also associated with progressive atrophy and extensive neuronal death in the temporal lobe, hippocampus, frontal cortex, and other brain areas [50,51]. Alzheimer’s disease develops from the accumulation of beta-amyloid (Aβ) and neuronal tangles comprising hyperphosphorylated tau proteins within neural progenitor fibers. Subsequent neurodegeneration and microglial activation mediate neuroinflammation in the brain [52,53,54,55,56,57]. Oxidative stress, neuroinflammation, and Ca^2+^ overload are significant in the development of Alzheimer’s disease. NLRP inflammasome promotes oxidative stress and inflammation in the brain [58]. 

There is a significant link between NLRP inflammasome and the pathogenesis of Alzheimer’s disease [59,60]. Aβ plaques not only induce oxidative stress and damage neurons, but also activate NLRP3 inflammasomes further releasing IL-1β to trigger neuroinflammation in patients with Alzheimer’s disease [61]. In neurons, NLRP1 inflammasome levels increase by approximately 25–30 fold in the patients with Alzheimer’s disease [62]. Neurotoxic effects of Aβ open the cellular ion channels, causing the inward and outward flow of calcium and potassium. An imbalance of K^+^/Ca^2+^ activates NLRP1 inflammasome in neurons, hence upregulating the expression of NLRP1 inflammasome, caspase-1, and IL-1β in LPS-primed macrophages [13]. Additionally, chronic glucocorticoid exposure is associated with neuronal degeneration, subsequently accelerating the deleterious progression in mice. Chronic glucocorticoid exposure in mice activates NLRP1 inflammasome-signaling pathways [13,63]. NLRP1 inflammasome inhibition by Ginsenoside Rg1 suppressed chronic glucocorticoid exposure-induced neuronal degenerations in mice [30,31]. Recent studies indicate that pyroptosis could also promote the development of Alzheimer’s disease. Chronic Aβ treatment significantly decreased PC12 cell viabilities and activated NLRP-1/caspase-1/GSDMD pathways, which was followed by the increased extracellular release of IL-18 and IL-1β [64].

NLRP3 inflammasome plays a role in destructive inflammatory responses by producing active forms of inflammatory cytokines. NLRP3 inflammasome activation modulates neuroinflammation, tissue damage, and cognitive impairment commonly found in the mouse model of Alzheimer’s disease [52]. Intestinal bacteria in the patients with Alzheimer’s disease mediate neuroinflammation via NLRP3 inflammasome activation [65]. Deposition and aggregation of Aβ in the brain of APP/PS1 mice stimulate NLRP3 inflammation, caspase-1, and IL-Iβ. In APP/PS1 mice, the process appears to be critical for the development of neuroinflammatory responses. In contrast, improved cognition using the Morris water maze (MWM) model and increased Aβ level was observed in NLRP3-knockout mice [66]. Primary microglia stimulation in vitro by fibrillar Aβ activates the production of NLRP3 inflammasome, caspase-1, causing the increased secretion of IL-1β in the animal models of Alzheimer’s disease [37]. Additionally, autophagy regulates the Aβ-induced activation of NLRP3 inflammasome via the LRP1/AMPK and the AMPK/mTOR/ULK1 pathway in Aβ-induced BV-2 cells and APP/PS1 mice [67,68]. Impaired autophagic processes in the microglia cause dysregulated Aβ clearance and severe deposition via the PRKAA1 pathway, potentially exacerbating inflammatory responses and NLRP3 inflammasome activation in mice [66]. The intraperitoneal injection of NLRP3 inflammasome inhibitor (JC-124) significantly improves the Aβ load in the brain of mice, suppressing neuroinflammatory responses and thereby producing neuroprotective effects [32]. A small molecule, MCC950, can inhibit NLRP3 inflammasome, repressing IL-1β and IL-18 secretion in mice [69]. The abnormal expression of tau proteins activates NLRP3 inflammasome, causing the subsequent release of IL-1β from the microglia [52,70,71]. NLRP3 inflammasome induces tau protein aggregation and hyperphosphorylation, promoting neuronal degeneration in mice [72,73,74]. Considering the non-negligible role of NLRP3/ASC/caspase-1 axis-mediated inflammation in Alzheimer’s disease, Alzheimer’s disease transgenic mice with selective suppression of NLRP3 inflammasome or caspase-1 expressions in the brain revealed significantly improvement cognitive functions [75]. The total Aβ volume significantly decreases in the hippocampus and cortex in NLRP3 or caspase-1-knockdown mice. Therefore, NLRP3/ASC/caspase-1 signaling pathways are implicated in the neuroinflammatory effects of Alzheimer’s disease [52,76,77,78].

Rats exhibited an improvement in spatial learning when treated with an anti-inflammatory drug (probenecid), which reduced NLRP1 inflammasome activation-mediated IL-1β and IL-18 secretion [38,39]. NLRP3 inflammasome inhibitors (CRID3) reduced tau hyperphosphorylation and aggregation by regulating tau kinase and phosphatase, improving spatial memory deficits in mice with Alzheimer’s symptoms [33,35]. Rats displayed significantly improved spatial memory after treatment with phosphatidylcholine (EPA-PC), inhibiting Aβ-induced toxicity by reducing NLRP3 inflammasome activation and increasing autophagy [79]. Osthole (OST) reduced hippocampal Aβ deposition and improved cognitive dysfunctions in the rat model of Alzheimer’s disease by NLRP3 inflammasome suppression via a PI3K/Akt/GSK-3β signaling pathway [36,80]. NLRP inflammasome is a potential treatment target for the progression of Alzheimer’s disease.

### 3.2. Parkinson’s Disease

Parkinson’s disease is one of the most common neurodegenerative disorders and the classical motor symptoms include bradykinesia, tremors, rigid movements, and dementia [81]. One of the primary causes of Parkinson’s disease is a gradual loss of dopaminergic (DA) neurons in the substantial nigra pars compacta (SNPC) [82]. Lewy bodies (LB) formation primarily comprising fibrillar alpha-synuclein (a-Syn) is also evident in patients with Parkinson’s disease [83]. Peripheral immune cell infiltrations and the activation of microglia and astrocytes have been reported in patients with Parkinson’s disease. These changes cause neuroinflammation [84,85]. Central and peripheral inflammation occurs in the prodromal stage of Parkinson’s disease, and remains sustained throughout the disease’s progression [81]. The abnormal regulation of a-Syn activates microglia to produce inflammatory factors and damage neurons in mice models of Parkinson’s disease [81]. Production of these cytokines is primarily regulated by the nuclear factor kappa B (NF-kB) and multi-protein inflammasome complexes, including NLRP1 inflammasome, NLRP3 inflammasome, and caspase-1 [86]. A clinical study collected serum samples from 12 untreated patients with Parkinson’s disease (aged 63–78 years) and detected a significantly upregulated expression of NLRP3 inflammasome, caspase-1, and IL-1β levels [87]. In BV2 cells, α-Syn activates NLRP3 inflammasome, followed by the release of caspase-1 and IL-1β. Increased caspase-1 and IL-1β levels cause neuroinflammation, subsequently damaging dopaminergic neurons [88,89,90,91,92,93,94,95,96]. The inhibition of the hepatic Nlrp3 protects dopaminergic neurons by attenuating systemic inflammation in a MPTP/p mouse model of Parkinson’s disease [97]. Based on recent studies, the inflammasome spontaneously assembles in mice and human DA neurons when parkin function is lost, which causes DA neuron death and the symptoms of Parkinson’s disease in the animals [98].

NLRP3 inflammasome-mediated neuroinflammation exerts a significant effect on the pathogenesis of Parkinson’s disease. NLRP3 inflammasome inhibitors promote the survival of DA neurons. NLRP3 inflammasome blockade significantly prevents α-syn-induced microglial activation and IL-1β production, preventing neuronal damage of midbrain DA neurons, ultimately improving the symptoms in patients with Parkinson’s disease [99]. Ellagic acid (EA) harbors profound implications in protecting DA neurons by inhibiting NLRP3 inflammasome activation in the microglia [40]. Safflower flavonoid extract (SAFE) reduced the level of plasma inflammatory factors, inhibiting NLRP3 inflammasome activation in mice, exhibiting significant anti-Parkinson’s disease effects [41]. Naringin (NAR) inhibited microglial NLRP3 inflammasome signaling activation and pro-inflammatory factor release in rats and protected DA neuron viabilities in rats [42]. Echinacoside (ECH) promoted the survival of DA neurons and inhibited microglial-mediated NLRP3/Caspase1/IL-Iβ inflammatory signaling pathway activation in the substantia nigra (SN) in mice [43]. These results highlight the neuroprotective effects of NLRP3 inflammasome inhibitors in the occurrence and progression of Parkinson’s disease [43].

### 3.3. Huntington’s Disease 

Huntington’s disease is an autosomal dominant neurodegenerative disease caused by expansions of triplet repeats encoded by polyglutamine sequences in the N-terminal region of the proteins associated with Huntington’s disease (mHTTs). Moreover, Huntington’s disease is characterized by impaired motor and cognitive functions, brain atrophy, weight loss, and reduced life expectancy [100].

Aggregation of mutant huntingtin culminates in neuronal dysfunction and death in patients with Huntington’s disease. In patients with Huntington’s disease and mice modeling Huntington’s disease, mHTTs were highly expressed in both neurons and microglia, whereas mutated mHTT aggregate to form inclusion bodies. Mutated mHTT expression activates NLRP3 inflammasome, causing neuroinflammatory responses, disrupting brain cell functions, and causing ultimate neuronal dysfunctions or even death [97]. Unlike in healthy individuals, NLRP3 inflammasome levels significantly increase in peripheral blood mononuclear cells (PBMCs) in patients with Huntington’s disease [101]. Galectin (Gal), which includes the carbohydrate recognition structural domain of β-galactosidase, affects the development of comorbid cognitive impairment in patients with Huntington’s disease. Gal3 damaged neurons by improving the secretion of NLRP3 inflammasome and IL-1β in the mice model of Huntington’s disease [97]. R6/2 mice (transgenic mice of Huntington’s disease) with Gal3 gene mutation in the striatum reduced the number of mHTT and inhibited NLRP3 inflammasome activation in the microglia, mitigating the effects of neuronal damage [97,102].

NLRP3 inflammasome inhibition or the use of other immunosuppressive agents reduces the pathophysiological changes in Huntington’s disease. PAP (papaverine) suppresses NLRP3 inflammasome activation by regulating NF-κB and CREB signaling pathways in mice, hence inhibiting microglial activation and neuronal cell death [45]. Tail vein injections of LV3-siNLRP3 in mice suppress hepatic pro-inflammatory cytokine production, down-regulate hepatic NLRP3 protein expression, and inhibit NLRP3 inflammasome activation, subsequently alleviating midbrain DA neuronal damage [103]. MIF (macrophage migration inhibitory factor) reduces the expression of NLRP3 inflammasome and inflammatory factors induced by MPP+ (1-methyl-4-phenylpyridinium) in microglia, reducing DA neuronal damage and exerting protective effects against neuroinflammation induced by Parkinson’s disease [104]. NLRP3 inflammasome inhibition by MCC950 decreases microglial inflammatory vesicle activation in mice, protects DA neurons in the substantia nigra, and suppresses motor dysfunctions in the mice model of Huntington’s disease [94,105]. RRx-001, the NLRP3 inflammasome inhibitors, reduces IL-1β and IL-18 expressions, inducing a decrease in T cell initiation and T cell trafficking to the brain and improving the course of experimental allergic encephalomyelitis (EAE) [46,47].

### 3.4. Depression

Abnormalities in cytokines and innate immunity receptors, including NLRP inflammasome, have been observed in the postmortem brains of depressed individuals. The protein and mRNA expressions of NLRP1, NLRP3, NLRP6 inflammasome, caspase-3, and ASC are significantly increased in individuals with major depression in the prefrontal cortex [106]. Since proinflammatory cytokine levels in the serum of people with major depression were significantly increased [106], inflammatory abnormalities have been involved in the pathophysiology of depression. NLRP1 inflammasome is significantly activated undergoing chronic stress in the hippocampus of rats [38,107]. NLRP1 inflammasome inhibition attenuates the depression-like behaviors and inhibits the secretion of mature IL-1β in the hippocampus of rats via the PKR/NLRP1 inflammasome pathway [107].

### 3.5. Drug Use Disorder

Drug use disorder is a chronic brain disorder with devastating consequences for individuals and society [108]. Drugs interact with the neuroimmune system to change neuroimmune gene expression and signaling, resulting in neurotoxicity. Chronic drug exposure causes compulsive drug use behaviors and long-lasting cravings, along with severe cognitive dysfunctions [109]. 

Neuroinflammation is a major underlying mechanism of methamphetamine (METH)-induced cognitive deficits. Increased levels of hippocampal NLRP1 and NLRP3 inflammasome expression as well as the induction of inflammation and apoptosis were found in 11 patients with METH drug use disorder [110]. METH promotes NLRP inflammasome release, and upregulates caspase-1 expressions, causing the aggregation of apoptosis-associated ASC proteins from rat cortical microglial. These outcomes are followed by the maturation and secretion of IL-Iβ, eventually resulting in neuroinflammation and neurotoxicity [34,111]. METH causes toxic effects in primary rat striatal neurons, cortical neurons, and PC12 cells, resulting in apoptosis and autophagy. The toxic effects of METH were accompanied by a significant increase in NLRP1 inflammasome expression, suppressed by knocking down the NLRP1 inflammasome gene [112]. METH further causes microglia to participate in neuroinflammatory effects by activating NLRP inflammasome [113]. In BV2 cells, NLRP3 inflammasome blockade by MCC950 significantly inhibits METH-induced increases in iNOS (a marker of activated microglia) expression, reducing microglial activation and cytotoxicity [34]. Mice treated with intraperitoneal injections of METH showed increased NLRP3 inflammasome levels and caspase-1 activation in the hippocampal brain regions [114]. NLRP1 inflammasome and downstream NLRP1/Caspase-1/GSDMD signaling pathways have important roles in the METH-induced cognitive function in rats. Using the new object recognition test, METH induced significant cognitive impairment and increased activity of NLRP1, cleaved-Caspase-11, IL-1β and TNF-α in the rat hippocampus. These phenomena were attenuated by aspirin-triggered-lipoxin A4 (ATL), a potent anti-inflammatory mediator [115]. In various cells, including cardiomyocytes, microglia, and neurons, METH induces apoptosis and pyroptosis through the NLRP-Caspase1-GSDMD pathway [116]. Therefore, METH exposures increased NLRP1 and NLRP3 inflammasome levels as well as neuroinflammation responses in the brain, followed by neurotoxicity and substantial damage to the brain.

Cocaine use disorder has been demonstrated to increase the level of oxidative stress and induce neuroinflammation which produces detrimental effects on the central nervous system in cocaine-disorder co-occurring AIDS patients [117]. Cocaine exposure was associated with the increased expression of various pro-inflammatory cytokines, NLRP1 inflammasome as well as adhesion molecules [118]. The expression of NLRP3 inflammasome in the cortical brain tissues of cocaine-dependent patients was significantly higher than that of the control participants [119]. Microglial activation mediated by cocaine is suggested to be involved with both ROS and NLRP3 inflammasome [120]. Studies using an NLRP3 blocker (MCC950) and siNLRP3 have also demonstrated the essential role of NLRP3 inflammasome in cocaine-mediated activation of inflammasomes and microglial activation in mice in both the striatum and the cortical regions [119]. NLRP3 inflammasome is a potential therapeutic target for relieving cocaine-mediated neuroinflammation.

Excessive ethanol consumption causes neurotoxicity via oxidative stress, inflammation, and cell death of the brain tissues in male C57BL/6 mice [121]. NLRP inflammasome is also involved in the neurotoxicity of alcohol [122]. Alcohol exposure significantly upregulates the expression of NLRP3 inflammasome and caspase-1. Alcohol increases caspase-1 and IL-1β expression in the central nervous system of wild-type mice, but not in NLRP3 or ASC knockout mice. Caspase-1 is subsequently activated, causing mitochondrial dysfunction in mice [123,124]. Previous studies have shown the role of NLRP3 inflammasome in alcohol-induced astrocyte inflammation and associated its activation with the production of ROS. This suggests that the activation of NLRP3 inflammatory in astrocytes is caused by alcohol exposure, eventually exacerbating the accumulation of mitochondrial ROS and the occurrence of cell death [125,126,127]. This suggests a critical role for NLRP3 inflammasome in regulating the effects of alcohol-induced neuroinflammation and neurotoxicity [128]. Calcium overload by alcohol has been shown to promote NLRP1, and NLRP3 inflammasome formation in rat cortical neurons and human neuroblastoma cells via the CaMKII/JNK1 pathway [129,130]. ROCK2 inhibition decreases the expression of NLRP3 inflammasome in astrocytes and attenuates alcohol-induced neuronal damage. This shows that ROCK2 downregulation suppresses the activation of NLRP3 inflammatory bodies [131]. Disulfiram (DSF), by reducing oxidative stress and blocking NLRP3 inflammasome, is an FDA-approved drug to treat chronic alcohol-use disorder. Additional studies indicate that DSF treatment alleviates the function of the lowered left heart and the apoptosis of myocardial cells by suppressing the activation of NLRP3 inflammasome in mice [48]. Considering the abstinence and cardiac protection impacts of DSF via the inhibition of NLRP inflammasome, the clinical application of NLRP inflammasome inhibitors represents a novel approach to protecting and treating disease-induced inflammation.

Significant NLRP3 inflammasome activation was observed in the prefrontal cortex and peripheral blood of morphine-treated mice [132]. Repeated exposures to morphine in wild-type mice increased the level of inflammation-related signals including NLRP3 inflammasome via the TLR4/NF-κB/NLRP3 pathway [133]. Procyanidins inhibit the morphine-induced activation of NLRP3 inflammasome and inflammatory responses in the microglia [134]. The different neuropsychiatric disorders and NLRP inflammasome inhibitors are summarized in Table 2.

## 4. Outlook

Many studies have demonstrated the basis for the identification of novel therapeutic approaches for neuropsychiatric disorders. Several studies presented in this review highlighted the importance of regulating NLRP inflammasome to delay the progression and outcomes of neuroinflammation in neuropsychiatric disorders. Clinical trials have also investigated the therapeutic effects of NLRP inflammasome inhibitors on alcohol-use disorder [135,136]. More clinical trials would be critically needed, in order to provide more solid evidence about the potential benefits of NLRP inhibitors. Furthermore, it appears that different psychiatric disorders share a common mechanism, neuroinflammation. The relationship between NLRP inflammasome and other neuropsychiatric disorders, including schizophrenia, autism, and obsessive-compulsive disorder, remains largely unknown. NLRP inflammasome inhibitors may potentially have a common therapeutic effect on different neuropsychiatric disorders.

## Figures and Tables

**Figure 1 brainsci-12-01057-f001:**
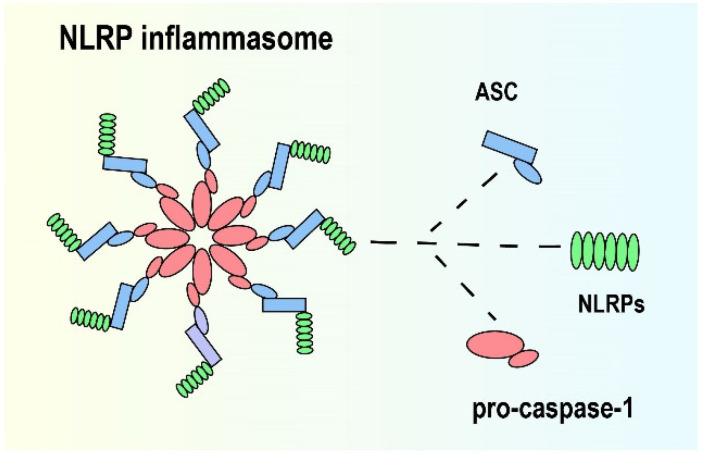
Structure of NLRP inflammasome.

**Figure 2 brainsci-12-01057-f002:**
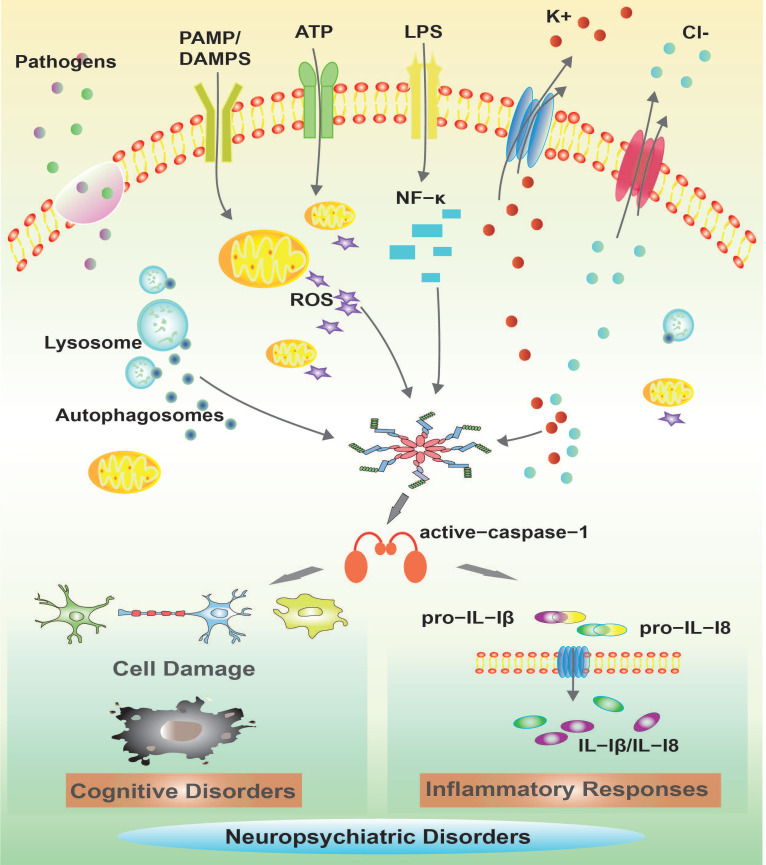
Mechanisms of NLRP inflammasomes in neuropsychiatric disorders.

**Table 1 brainsci-12-01057-t001:** Basic information on NLRP inflammasome inhibitors for the treatment of neuropsychiatric disorders.

Name	Structure	Model	Findings	Clinical Advance	References
Rg1	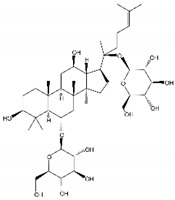	APP/PS1 mice	hippocampus and cortex of mice Aβ↓.		[30,31]
JC-124	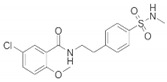	CRND8 APP transgenic mice (TgCRND8)	Aβ, Aβ1-42↓		[32,33,34]
MCC950 (CRID3)	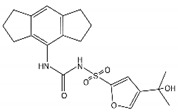	In vitro cell culture model of Alzheimer’s disease (SH-APP cells); METH use disorder.	caspase-1 and IL-1β levels, tau, Aβ↓; Capase-1, lysosomal histone B activity, ROS↓.		[35]
Osthole	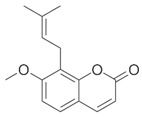	APP/PS1 double transfected mice	hippocampal tau↓		[36,37]
Probenecid	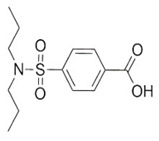	Aged rats	Inflammasome activation↓ spatial learning performance↑.		[38,39]
EA (Ellagic acid)	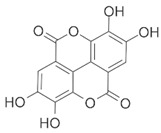	LPS induced rat DA neuronal (MN9D, BV-2 cell line) damage model. NLRP3 siRNA.	Neuroprotection.		[40]
SAFE (Safflower Flavonoid Extract)	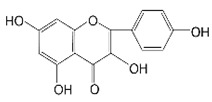	6-Hydroxydopamine (6-OHDA)-induced rat model of Parkinson’s disease	NLRP3 inflammasome activation ↓		[41]
NAR(Naringin)	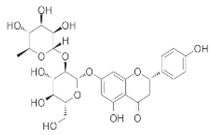	Rat nigral stereotaxic injection of lipopolysaccharide (LPS). NLRP3 siRNA.	Neuroprotection.		[42]
ECH(Echinacoside)	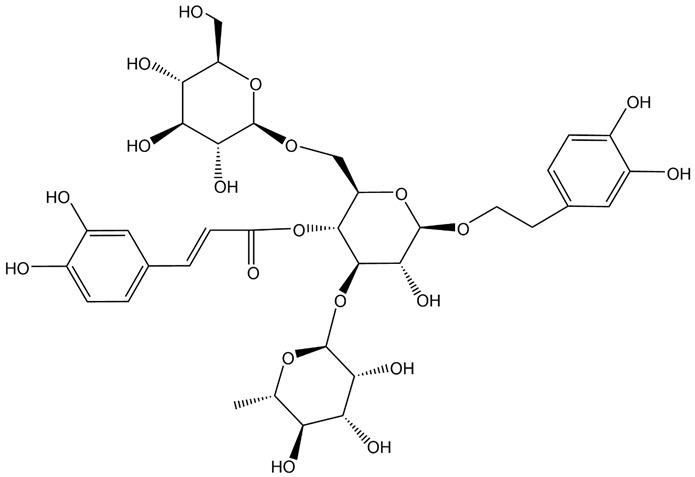	Mice model of Parkinson’s disease	NLRP3/Caspase-1/IL-1β signaling pathway↓.		[43]
Laquinimod	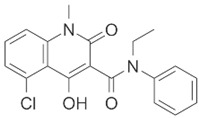	Parkinson’s disease			[44]
PAP(Papaverine)	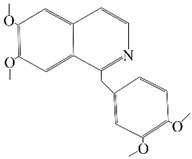	Mice model mice of Huntington’s disease	Regulation of NF-κB and CREB signaling pathways to inhibit NLRP3.		[45]
RRx-001	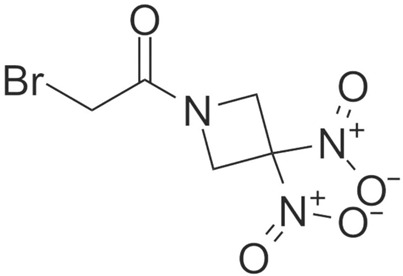	Hunting’s disease	NLRP3↓.		[46,47]
DSF(Disulfiram)	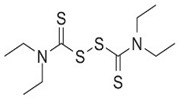	Alcohol use disorder	ROS and NLRP3↓	Alcohol use disorder	[48]
Cod liver oil	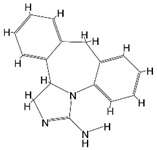		Alcohol consumption↓	Alcohol use disorder	[49]

**Table 2 brainsci-12-01057-t002:** Different neuropsychiatric disorders and NLRP inflammasome inhibitors.

Diseases		Drugs	Pathways	Pathological Changes	Performance	References
Alzheimer’s disease		Rg1, JC-124, CR2D3, EPA-PC, OST, Probenecid	K^+^/Ca^2+^-NLRP-Caspase1-IL-1β	Aβ deposition, tau protein entanglement.	Loss of memory function and cognitive decline.	[51,53,56,63,65]
Parkinson’s disease		EA, SAFE, NAR, ECH, Laquinimod	a-Syn-NLRP-Caspase1	Decrease in dopaminergic neurons and formation of Lewy bodies.	The Campaign Triad.	[83,89,95,97,100]
Huntington’s disease		PAP, MCC950	Gal3-NLRP3-IL-1β	Inclusion body formation, brain atrophy.	Motor, cognitive impairment.	[101,102,104,105]
Depression			caspase3-NLRP-IL-1β	Inflammatory lesion.	Persistent depression and loss of interest.	[109,110]
Drug use disorder	METH	MCC950	NLRP-Caspase-1-IL-1β	ASC protein aggregation and increased autophagosomes.	Cognitive dysfunction, schizophrenia.	[114,115,117,119]
	Cocaine	MCC950	ROS/NLRP	Oxidative stress and neuroinflammation.	Nervous and mental injury.	[120,123]
	Alcohol	Cod liver oil, VX765	CaMKII/JNK1-NLRP1	Lysosomal and mitochondrial damage.	Central nervous system lesions, behavioral disorders.	[125,127]

## Data Availability

Not applicable.

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
