# Peer review of "The Significance of NLRP Inflammasome in Neuropsychiatric Disorders"

_brainsci, 2022, doi:10.3390/brainsci12081057_

Round 1

Reviewer 1 Report

The review by Shen et al. reports on linkages between NLRP/inflammasome activation and neuropsychiatric disorders. The manuscript is clearly written and succinct while encompassing many important areas. There seems to be a decent consideration for the impacts of interventions targeting inflammasomes for the treatments of such disorders, whether they be bona fide inhibitors or dietary supplements that may inhibit inflammasomes or function through a myriad of additional mechanisms. Specific requests for modification / editing follow.

1.     In Section 1, explain the NLRP3 acronym upon its first usage (a NLR With a pyrin domain).

2.     In Section 2, line 61, explain the ASC acronym (apoptosis-associated speck-like protein containing a CARD (caspase activation and recruitment domain). Also, ASC is conventionally referred to as an “adaptor protein” rather than a “junction protein”.

3.     Delete “NLRP1 and NLRP3 inflammasome.” From line 64.

4.     In Figure 1, replace “PRR” with “NLRPs”.

5.     In Section 2, the authors should include information about the agonists that cause activation of NLRP1 and NLRP3, especially those that are particularly relevant to neuropsychiatric disorders.

6.     Also in Section 23, the authors should convey that myeloid cells within and outside the neuronal system, in addition to microglia, also contain NLRP1 and NLRP3, and that activation therein may also contribute to neuropsychiatric disorders.

7.     ON line 94, change “folds” to “fold”.

8.     On line 112, change “procedure” to “process”.

9.     On line 114, the authors cite reference # 36 to support the notion that NLRP3 knockout mice have improved cognitive function and increased Abeta levels in an Alzheimer’s Disease model. However, no NLRP3 knockout mice were used in reference #36. Should this instead be reference #45? It is STRONGLY recommended that all references be checked for accuracy.

10.  On line 132, change “improved” to “improvement”.

11.  On line 150, change PD remained” to “PD, and remain”.

12.  On line 156, reference #61 has nothing to do with the sentence written.

13.  Reference #68 is formatted incorrectly.

14.  On line 162, 163, 165, and 208, delete “The” from the beginning of these sentences.

15.  Throughout Section 3, spell-out the acronyms for the diseases discussed (AD, HP, and PD).

16.  On line 178, describe “HD”., It seems this should instead be Huntingtin Protein (HTT or mHTT).

17.  On line 184, change “expressions activate” to “expression activates”.

18.  On line 196, change “NF-kbeta” to “NF-kappaB” and make the kappa into the Greek symbol.

19.  On line 252, add an “of” between “level” and “oxidative”.

Reviewer 2 Report

In this submitted review article titled “The significance of NLRP inflammasome in neuropsychiatric disorders”, Shen et al presented here a summary of NLRP inflammasome activation mechanism in neuropsychiatric disorders including Alzheimer`s disease, Huntington`s disease, Parkinson`s disease, drug use disorders. The authors also briefly discussed the therapeutic potential of inflammasome activation inhibitors against these disorders. 

Overall, while this review provided seemingly comprehensive summary of the role of inflammasome in neuropsychiatric disorders, there are several points should be improved. 

Major points:

1: can the authors please provide some detailed signaling pathways that are involved in NLRP1/NLRP3 inflammasome activation, for example ROS elevation, how pattern recognition receptor is overexpressed, how K+/Ca2+ is imbalanced in these disorders…, I recommend the authors to summarize these mechanisms in Figure 2.

2: What is the clinical advance to treat these disorders for NLRP inflammasome inhibitors that are listed in Table 1?

3: Can the authors please provide some insights regarding how inflammasome-mediated neuropsychiatric disorders should be advanced in the near future?

Minor points:

1: Figure 1 and Figure 2 have some overlap, please consider removing the above portion in figure 2 and add what I mentioned in point 1 of major comments.

2: ASC stands for apoptosis-associated speck-like protein containing a CARD. In the text, there are several inappropriate names used for ASC, for example, line 236 (spot-like), Table 1 (speckle-like)…, please fix these.

3: The authors are confusing the readers the concepts of inflammasome and pattern recognition receptor NLRP1/3. In the manuscript, for example line 94, 249…, should these refer to pattern recognition receptor NLRP1/3?

4: what is the NLRP1 inhibitor in Table 1 (reference 90, 91)? I would also suggest the authors to draw the chemical structures of the drugs mentioned in Table 1. Maybe put in separate figure?

Reviewer 3 Report

The manuscript could provide important insight into the research of neuropsychiatric diseases. 

However, this is a mini-review. Please include more details of molecular mechanism, the pathways involved, cell line or animal models used in those studies, animated figures of them etc. for improving the manuscript.

Round 2

Reviewer 2 Report

In this revised manuscript, the authors have addressed most of comments. I recommend its publication in Brainsci and I have only some minor suggestions:

1: Can the author please add chemical structures that are missing in Table 1?

2: CRID3 and MCC950 are the same, please use on name instead. Also, please draw the structure in free form instead of sodium salt

3: there are two structures in DSF

4: Structure of ECH is two small to see

Author Response

Dear editor,      We thank both you and the reviewer for the review of our manuscript “The significance of NLRP inflammasome in neuropsychiatric disorders” (brainsci-1828649).

As suggested, we have revised the manuscript. Below is a point by point response to each of the comments by the reviewer.

Sincerely,

Yu

Response to Reviewer

Major points:

1: Can the author please add chemical structures that are missing in Table 1?

The chemical structures are now added in revised Table 1, as suggested.

2: CRID3 and MCC950 are the same, please use one name instead. Also, please draw the structure in free form instead of sodium salt

MCC950 is now used throughout the revised manuscript. The structure in free form instead of sodium salt is now used in the revised manuscript, as suggested.

3: there are two structures in DSF

The structure of DSF is now corrected, revised, as suggested.

4: Structure of ECH is too small to see

The structure of ECH is now enlarged. The original image of each structure is also included in a supplemental file.
